# Distinct Outcomes in COVID-19 Patients with Positive or Negative RT-PCR Test

**DOI:** 10.3390/v14020175

**Published:** 2022-01-18

**Authors:** Maria Clara Saad Menezes, Diego Vinicius Santinelli Pestana, Juliana Carvalho Ferreira, Carlos Roberto Ribeiro de Carvalho, Marcelo Consorti Felix, Izabel Oliva Marcilio, Katia Regina da Silva, Vilson Cobello Junior, Julio Flavio Marchini, Julio Cesar Alencar, Luz Marina Gomez Gomez, Denis Deratani Mauá, Heraldo Possolo Souza

**Affiliations:** 1Emergency Medicine Department, Faculdade de Medicina da Universidade de São Paulo, São Paulo 01246-903, Brazil; diego.pestana@fm.usp.br (D.V.S.P.); lgomez928@gmail.com (L.M.G.G.); heraldo.possolo@fm.usp.br (H.P.S.); 2Hospital das Clínicas, Faculdade de Medicina da Universidade de São Paulo, São Paulo 01246-903l, Brazil; juliana.ferreira@hc.fm.usp.br (J.C.F.); carlos.carvalho@hc.fm.usp.br (C.R.R.d.C.); marcelo.felix@hc.fm.usp.br (M.C.F.); izamarcilio@gmail.com (I.O.M.); katia.research@gmail.com (K.R.d.S.); vilson.cobello@hc.fm.usp.br (V.C.J.); julio.marchini@hc.fm.usp.br (J.F.M.); julio.alencar@hc.fm.usp.br (J.C.A.); 3Institute of Mathematics and Statistics, Universidade de São Paulo, São Paulo 05508-090, Brazil; denis.maua@usp.br

**Keywords:** SARS-CoV-2, COVID-19 testing, COVID-19, hospital mortality, intubation

## Abstract

Identification of the SARS-CoV-2 virus by RT-PCR from a nasopharyngeal swab sample is a common test for diagnosing COVID-19. However, some patients present clinical, laboratorial, and radiological evidence of COVID-19 infection with negative RT-PCR result(s). Thus, we assessed whether positive results were associated with intubation and mortality. This study was conducted in a Brazilian tertiary hospital from March to August of 2020. All patients had clinical, laboratory, and radiological diagnosis of COVID-19. They were divided into two groups: positive (+) RT-PCR group, with 2292 participants, and negative (−) RT-PCR group, with 706 participants. Patients with negative RT-PCR testing and an alternative most probable diagnosis were excluded from the study. The RT-PCR(+) group presented increased risk of intensive care unit (ICU) admission, mechanical ventilation, length of hospital stay, and 28-day mortality, when compared to the RT-PCR(−) group. A positive SARS-CoV-2 RT-PCR result was independently associated with intubation and 28 day in-hospital mortality. Accordingly, we concluded that patients with a COVID-19 diagnosis based on clinical data, despite a negative RT-PCR test from nasopharyngeal samples, presented more favorable outcomes than patients with positive RT-PCR test(s).

## 1. Introduction

COVID-19 patients usually present systemic and respiratory symptoms, such as fever, cough, and shortness of breath [1]. Radiological exams may show different degrees of lung involvement, and laboratory tests may show increased inflammatory markers [1,2,3]. None of these factors are specific, and the diagnosis is obtained only when the SARS-CoV-2 virus is detected in the airways [4].

During the initial times of the pandemic, there were concerns regarding the accuracy of the tests used [5]. A meta-analysis published in May 2020 found that the accuracy of the RT-PCR test for coronavirus diagnosis can change according to the prevalence of COVID-19. With a prevalence of 50%, common among health professionals with respiratory symptoms, a post-test probability of 96% was found. With a prevalence of 20%, the post-test probability was 84%. With a prevalence of 5%, they found a 55% post-test probability [6]. More recently, another meta-analysis showed that the test’s accuracy has improved over the year; however, a marked heterogeneity in the proportion of false-negative RT-PCR results amongst different tests is still maintained [7]. The heterogeneity is largely unexplained. There are several reasons that can underlie this heterogeneity. Researchers have suggested that these failures in SARS-CoV-2 detection are related to multiple preanalytical and analytical factors, such as lack of standardization for specimen collection, delays, or poor storage conditions before arrival in the laboratory, the use of inadequately validated assays, contamination during the procedure, insufficient viral specimens and load, the incubation period of the disease, and the presence of mutations that escape detection [7].

Nevertheless, even with improvements in virus detection, there is a group of patients diagnosed with COVID-19 based solely on clinical criteria because they do not present a positive RT-PCR test [8]. In this group of patients, the differential diagnosis for COVID-19 should be ruled out. This group of differential diseases includes mainly respiratory diseases, infections of other origins, cardiovascular, oncological, gastrointestinal, urogenital, and neurological diseases [9].

Clinical criteria for diagnosing COVID-19 include the initial assessment of related symptoms and exposure history [2], coupled with typical laboratory findings (lymphopenia, thrombocytopenia, and elevated liver enzymes, lactate dehydrogenase, inflammatory markers, and D-dimer) [3], and characteristic images at the lung CT scan (multiple bilateral ground-glass opacities in the peripheral lower lung zones) [10].

These presumptive diagnostic criteria have been reported to be more sensitive than the RT-PCR, providing a positive result at earlier stages of the disease [11]. Currently, both methods are described as complementary and should be used in conjunction when available. 

Notwithstanding, it is unclear whether patients with a presumptive diagnosis of COVID-19 evolve similarly to patients who were diagnosed by the RT-PCR test. Therefore, in this study, we explored the specificities of the aforementioned group by comparing 2292 COVID-19 hospitalized patients confirmed by SARS-CoV-2 RT-PCR (COVID-19 RT-PCR(+) group) with 706 COVID-19 hospitalized patients diagnosed by presumptive clinical criteria with negative SARS-CoV-2 RT-PCR results (COVID-19 RT-PCR(−) group). 

## 2. Materials and Methods

We conducted a retrospective unicentric cohort study from March to August 2020 at the Hospital das Clínicas da Universidade de São Paulo (HC-FMUSP), a 2200-bed urban, academic medical center comprising five institutes and two auxiliary hospitals. During the pandemic, the HC-FMUSP has been designated for the reception and care of patients with severe COVID-19. All patients were evaluated and treated according to the Institution’s protocol that consisted and still consists of supportive care, including supplemental oxygen and mechanical ventilatory support when indicated. By the end of July 2020, after the preliminary report from the RECOVERY Collaborative Group, patients who were receiving either invasive mechanical ventilation or oxygen alone were also treated with corticosteroids both in the COVID-19 RT-PCR(+) group and COVID-19 RT-PCR(−) group. Therefore, there were no differences in the treatment provided for both groups. There were no instances during the time-period studied when mechanical ventilation or other treatment modalities were unavailable for patients who might have needed them.

In this research study, the COVID-19 RT-PCR(+) group is formed by patients with a high clinical suspicion of COVID-19 and a positive RT-PCR test for the SARS-CoV-2 whereas the COVID-19 RT-PCR(−) group is formed by patients with a high clinical suspicion of COVID-19, judged by 2 experienced attending physicians after ruling out differential diagnosis, and at least two negative RT-PCR tests for the SARS-CoV-2.

### 2.1. COVID-19 Diagnostics 

Individuals with clinical suspicion of severe COVID-19 were referred to our tertiary hospital from other healthcare institutions in our city when the physician had a suspicion for COVID-19. All subjects had some sort of respiratory symptoms (cough or dyspnea) and diffuse infiltrates on X-rays.

Upon arrival, these patients were classified according to the RT-PCR test results for SARS-CoV-2 detection. Patients that arrived with a previous nasopharyngeal swab RT-PCR test result performed at the origin hospital were immediately included in the COVID-19 RT-PCR(+) group.

Patients with no previous test or a negative test were then submitted to the RT-PCR test at hospital admission. Patients with positive RT-PCR results were then allocated to the COVID-19 RT-PCR(+) group.

Patients with negative tests were then submitted to lung CT scans, laboratory tests, and another nasopharyngeal swab collection. When a diagnosis other than COVID-19 was made, the subject was excluded from our study. If the patient had more than 7 days of COVID-19 symptoms on hospital admission, COVID-19 serology was also performed. During the inpatient stay, patients that had at least two negative RT-PCR tests but maintained high clinical suspicion for COVID-19 also underwent serological testing after, at least, 7 days of symptoms. Unfortunately, due to a shortage of resources, serologic COVID-19 testing could not be performed in all eligible patients. COVID-19 serology at discharge was not performed. In every serological testing occasion, both IgM and IgG were tested, and a positive result in either one was considered enough to consider that patients had positive serology status.

At discharge, patients with no positive SARS-CoV-2 RT-PCR result but who were considered by sequential attending physicians (at least 2) with clinical and radiological suspicion of COVID-19, after ruling out differential diagnoses, were allocated to the COVID-19 RT-PCR(−) group. Figure 1 illustrates the institutional testing protocol and the distribution of participants. Of note, CT scans were reported by specialist radiologists, and the results were freely available to those who adjudicated on the presence or absence of COVID-19.

Therefore, the RT-PCR(+) group was composed of patients with clinical, laboratory, and radiological findings of COVID-19 and a positive nasopharyngeal swab for SARS-CoV-2 at some point of the disease evolution. The RT-PCR(−) group was composed of patients with clinical, laboratory, and radiological findings of COVID-19, other differential diagnoses excluded, and no positive RT-PCR test for SARS-CoV-2 during their hospital treatment. 

Patients with a differential diagnosis more probable than COVID-19 were excluded as well as patients younger than 18 years of age. Before hospital discharge, all patients were assessed for the presence of antibodies against SARS-CoV-2 in circulating blood. Routine screening for other respiratory viruses was undertaken whenever there was uncertainty regarding the COVID-19 diagnosis. Patients with a positive result for another virus were excluded from this study.

### 2.2. Data Collection 

We designed an extensive database to characterize all hospital admissions for COVID-19 from March to August 2020. Datasets (demographics, comorbidities, clinical conditions, treatment, laboratory tests, and outcomes) derived from the electronic health record were directly imported to the Research Electronic Data Capture (REDCap) [12] system hosted at Hospital das Clínicas da Faculdade de Medicina da Universidade de São Paulo. These datasets were submitted to a rigorous process of harmonization and consolidation before being imported to REDCap. In addition, a task force of researchers reviewed the medical records to ensure data completeness and quality. 

### 2.3. RT-PCR Assessment 

The patient’s nasopharyngeal or oropharyngeal secretions, or both, were collected via swab. An automated separation kit was used to extract and purify viral RNA from samples using magnetic microparticles (Abbott mSample Preparation Kit RNA, Des Plaines, IL, USA). The reverse transcription, amplification, and real-time detection protocols were validated by our institutional Laboratory Division (accredited by the College of American Pathologists). First- and second-line screening tools, E gene assay and N gene assay, respectively, were used for confirmation, as described by Corman et al. [13]. Extraction and amplification were internally controlled in every sample using the endogenous gene RNAseP, besides positive and negative controls in all batches. The analytical sensitivity was 40 RNA copies/mL, and specificity was 100% even in samples containing other respiratory viruses. Of note, some patients were transferred from smaller facilities to our institution and were tested for SARS-CoV-2 infection before admission in our hospital; tracking the exact kit and reagents used for testing this subset of patients was not feasible. 

The Abbott mSample Preparation Kit RNA has been authorized by the United States of America Food and Drugs Administration (FDA) as of July 2020 and is currently in accordance with the latest recommendations made by the Center of Disease Control and Prevention (CDC) regarding the use of nucleic acid amplification tests for diagnosing SARS-CoV-2.

### 2.4. Statistical Analysis 

Baseline characteristics and outcomes of hospitalized patients in the COVID-19 RT-PCR(+) group and COVID-19 RT-PCR(−) group were compared using the Mann–Whitney test. Continuous variables were represented with medians, and interquartile ranges and categorical variables were represented as proportions. A 2-sided *p* value of ≤0.05 was used to designate statistical significance. 

The survival probability and probability of not being submitted to intubation in the RT-PCR(+) group and RT-PCR(−) were calculated using the Kaplan–Meier method. Hazard ratio of the risk variables were estimated by fitting a Cox proportional hazards model. Afterwards, baseline factors that were associated with 28-day in-hospital mortality and intubation were identified using univariate logistic regression models. All variables that were statistically significantly associated with each outcome were then entered into separate multivariate logistic regression models. Adjusted odds ratios of mortality and intubation were calculated for each of these variables with 95% confidence intervals (CIs). Analyses were conducted using the Python frameworks statsmodels (v. 0.12.2), scikit-learn (v. 0.24.2) and lifelines (v. 0.26). Eventual missing data were set to be managed according to their nature. Values missed completely at random were addressed using complete case analysis; values missing at random were addressed using multiple imputation methods; values likely to be missing not at random will be re-evaluated by authors.

## 3. Results

### 3.1. Patients’ Characteristics and Presentation 

A total of 2998 patients were included in this study. Groups were divided according to their SARS-CoV-2 RT-PCR testing status, as described above. A total of seven hundred and six (23.5%) patients were allocated to the COVID-19 RT-PCR(−) group, and two thousand two hundred and ninety-two (76.5%) patients were allocated to the COVID-19 RT-PCR(+) group. A total of five hundred and seventy-seven patients were excluded from the study for having a more likely alternative diagnosis other than COVID-19. 

Demographic variables were similar between the two experimental groups, as shown in Table 1. Regarding comorbidities, a higher prevalence of diabetes, cardiovascular diseases, chronic kidney disease, peripheral vascular disease, and cancer was observed in the RT-PCR(+) group. Regarding clinical presentation at hospital admission, patients in the RT-PCR(−) group had a delay in seeking medical attention when compared to patients in the RT-PCR(+) group (Table 2). The median time after initial symptoms was eight days (IQR 5–11) in the former group, compared to seven days (IQR 4–10) in the latter (*p* < 0.001, 95% CI). 

An analysis comparing the days of COVID-19 symptoms on hospital admission among different comorbidities was performed to investigate whether the different rates of comorbidities between COVID-19 RT-PCR(+) group and COVID-19 RT-PCR(−) group were due to patients with a certain comorbidity arriving earlier to the hospital, as the probability of a positive test increases with the decrease in days of symptoms on hospital admission [14]. Median days of COVID-19 symptoms on hospital admission differed in some subgroups, according to certain baseline diseases: Diabetes vs. No Diabetes (7 and 7, *p* = 0.54, 95% CI Figure 1), Cancer vs. No Cancer (4 and 8, *p* < 0.001, 95% CI Figure 1), Hypertension vs. No Hypertension (7 and 7, *p* = 0.26, 95% CI Figure 1), Asthma vs. No Asthma (7 and 7, *p* = 0.46, 95% CI Figure 1), Chronic Kidney Disease vs. No Chronic Kidney Disease (6 and 7, *p* < 0.001, 95% CI Figure 1) and Cardiovascular Disease vs. No Cardiovascular Disease (7 and 7, *p* < 0.001, 95% CI Figure 1). 

Patients with negative RT-PCR presented more dyspnea (82.2% vs. 77.2%) and gastrointestinal symptoms than patients with positive RT-PCR (Table 2). All other symptoms were equal between the two groups. Physical examination between the two groups was similar, with a small but significant difference in oxygen saturation (Table 3). The median of oxygen saturation in the COVID-19 RT-PCR(+) group and COVID-19 RT-PCR(−) group was 94%, but the distribution of values was slightly different with lower oxygen saturation in the COVID-19 RT-PCR(-) group (*p* = 0.046, 95% CI).The SAPS score, which estimates the probability of mortality for ICU patients, was comparable in both groups. 

### 3.2. Laboratory Tests

Patients in the RT-PCR(−) group had a higher number of circulating leukocytes, lymphocytes, and D-dimer, and lower CRP values (Table 4). The presence of antibodies against the SARS-CoV-2 was detectable in a higher percentage of patients in the RT-PCR(+) group compared to the RT-PCR(−) group (88.4% vs. 78.6%, *p* < 0.001, 95% CI, Table 4. Altogether, the data presented above suggest that both experimental groups are similar in disease severity; however, some differences in previous diseases’ existence and laboratory tests were detected. As an attempt to establish baseline COVID-19 seropositivity in São Paulo, we checked the official data report released periodically by the mayor’s office. It was found that the seroprevalence in this population was 11.4% as of June 2020 and 13.3% as of August 2020. Data related to previous months were unavailable in the official records of São Paulo state administration [15]. As the majority of our cohort was enrolled before June 2020, less than 11.4% of included patients were expected to have a baseline positive COVID-19 serology.

Patients in the RT-PCR(+) group were admitted to ICU at higher rates (67.1% vs. 50.0% (*p* < 0.001, 95% CI Table 1)), were submitted to endotracheal intubation more frequently (55.9% vs. 40.5% (*p* < 0.001, 95% CI Table 1)), and stayed longer in hospital (13 days vs. 9 days (*p* < 0.001, 95% CI Table 1)) when compared to patients in the RT-PCR(−) group (Table 5). Moreover, the in-hospital and 28-day mortality rates were higher in patients with positive RT-PCR tests (*p* < 0.001, 95% CI).

The COVID-19 RT-PCR(+) group, when compared to the RT-PCR(−) group had a hazard ratio (HR) of 1.44 (95% CI 1.26–1.65, *p* < 0.005) for intubation (Figure 2) whereas for 28-day in-hospital mortality the HR was 1.09 (95% confidence interval (CI) 0.91–1.31, *p =* 0.33) (Figure 3). In a multivariate model adjusted for age, sex, smoking status, hypertension, diabetes, asthma, and cancer, a positive RT-PCR was independently associated with increased risk of intubation (adjusted odds ratio (aOR), 2.04; 95% CI, 1.70–2.44; *p* < 0.001, Table 2) when compared to patients with a negative RT-PCR. Additionally, in a multivariate model adjusted for age, sex, body mass index, smoking status, hypertension, cardiovascular disease, chronic obstructive disease, chronic kidney disease, and cancer, the risk of a 28-day in-hospital mortal was higher in patients in the COVID-19 RT-PCR(+) group compared to the COVID-19 RT-PCR(−) group (aOR, 1.75; 95% CI, 1.41–2.16; *p* < 0.001, Table 6).

An analysis of a subgroup of patients without comorbidities that differed between the COVID-19 RT-PCR(+) and COVID-19 RT-PCR(−) was also performed. Accordingly, only patients without cardiovascular disease, diabetes, chronic kidney disease, peripheral vascular disease, cancer, and hematologic malignancy were enrolled. In this subgroup, the positive RT-PCR association with intubation (64.4% vs. 47.6%, *p* < 0.001, Table 7) and 28-day mortality (30.8% vs. 19.3%, *p =* 0.002, Table 7) was maintained.

Importantly, only missing data completely at random were observed. Missing data at random and not at random were not observed. A strict evaluation of the dataset highlighted that missing data occurred due to insufficient filling of electronic medical records in punctual episodes. Accordingly, complete case analysis was used to minimize the biases derived from missing data.

## 4. Discussion

In this study, COVID-19 patients with at least one positive SARS-CoV-2 RT-PCR result were more likely to be intubated or die during hospitalization when compared to patients that had only clinical and radiological criteria for COVID-19 diagnosis. This association persisted even after an adjustment for age and comorbidities was performed.

The first question to be raised is whether participants in the RT-PCR(−) group really had COVID-19. This issue was approached by checking the antibody production in these patients. Although we did not have data from all patients, some patients (*n* = 351 patients) were tested and 78.6% presented serologic conversion after hospital admission. Even though this rate is lower than the one found in the RT-PCR(+) group, it is significant. This is indirect evidence that the clinical and radiological presumptive diagnosis was probably accurate in most patients in the COVID-19 RT-PCR(−) group. These data are in accordance with other authors showing that CT scan is more sensitive than RT-PCR for COVID-19 diagnosis [16] as RT-PCR test accuracy may be questioned. A total of five studies, involving 957 patients, demonstrated that false negative results range from 2 to 29% (equating to sensitivity of 71–98%) [7]. Although the authors judged their evidence as low, due to the risk of bias, indirectness, and inconsistency issues, the occurrence of false negative SARS-CoV-2 RT-PCR results must be considered. 

There are several reasons that can underlie false-negative RT-PCR nasopharyngeal SARS-CoV-2 results, including preanalytical steps (conservation of samples, time until being sent to the laboratory, and training of personnel) [17], the number of additional RT-PCR assays performed [7], and insufficient viral load (possibly influenced by time from the onset of symptoms) [13]. In our institution, the protocol used to test for SARS-CoV-2 infection requires at least 40 viral RNA copies per milliliter to detect viral presence in 95% of cases (defined as limit of detection (LOD)) suggesting that viral load may be one of the factors contributing to the overall number of false negative tests in this cohort. 

Likewise, another interesting aspect of this study is the relationship between the high number of participants in the RT-PCR(−) group and the RT-PCR protocol used. Corman et al. [13] suggested that the first-line protocol should include an RNA analysis of the viral gene E followed by an analysis of the viral gene RdRp. Alternatively, viral gene N could also be used instead of gene RdRp, although it had a higher LOD. In the present study, as it may occur in several other health care institutions throughout other BRICS’ nations, limitation of resources and logistics prevented the laboratory facility from using gene RdRp in the RT-PCR protocol. Due to the difference between the LODs, one may hypothesize that the protocol of using gene N contributed to the occurrence of false negatives, increasing the number of participants in the RT-PCR(−) group. However, as mentioned above, we believe that this potential limitation may be present in other institutions, which grants relevance to the presenting results as it depicts real, daily-care difficulties, and may guide practitioners during clinical decision-making in the context of limited hospital and laboratorial resources (e.g., ICU beds, mechanical ventilation, and updated diagnostic protocols, etc.). 

Patients with cardiovascular disease, peripheral vascular disease, diabetes, chronic kidney disease, active cancer, and hematologic malignancy were more frequent in the COVID-19 RT-PCR(+) group. A possible explanation for the higher rate of chronic comorbidities in the COVID-19 RT-PCR(+) group is that those patients had a more severe spectrum of COVID-19 [18] and presented earlier to the emergency department, which could have led to a higher chance of having a positive SARS-CoV-2 RT-PCR test, as the probability of a positive test increases with the decrease in days of symptoms on hospital admission [13]. This may be a feasible explanation for the higher cancer rate in the COVID-19 RT-PCR(+) group (13.1% vs. 8.8%, *p =* 0.004, Table 1) as the median days of symptoms on hospital admission for cancer and non-cancer patients are 4 and 8, respectively (Figure 1). However, this significant difference of median days of symptoms on hospital admission is not seen among other comorbidities (Figure 1). Additionally, the higher rate of comorbidities in the COVID-19 RT-PCR(+) group could explain the higher risk of intubation and mortality in that group, but the overall association between a positive SARS-CoV-2 RT-PCR and 28-day mortality and intubation persisted even after an adjustment for comorbidities, suggesting an independent association between SARS-CoV-2 RT-PCR and COVID-19 outcomes (Table 2). We are aware that unmeasured imbalances can persist after a statistical adjustment, especially in observational studies. 

Another hypothetical explanation for the higher rate of some comorbidities in the COVID-19 RT-PCR(+) group is a systemic difficulty to clear the SARS-CoV-2 viral load. Our results are in line with previous evidence, which demonstrated that COVID-19 patients with cardiovascular disease, chronic kidney disease, active cancer, and hematologic malignancy present a higher viral load on hospital admission and, thus, a higher probability of a positive RT-PCR test [19]. Reasons for higher viral loads specifically in these populations are not fully understood and warrant further investigation. However, in previous studies, chronic obstructive pulmonary disease and age were also associated with higher viral load [19] whereas in the present study they were not different among COVID-19 RT-PCR(+) and COVID-19 RT-PCR(−) groups. This may be caused by the limitations of evaluating only SARS-CoV-2 RT-PCR results. 

Although the COVID-19 RT-PCR(+) group had a higher rate of comorbidities, after an adjustment for the factors that are associated with a worse COVID-19 outcome, the aforementioned group maintained an excessive incidence of intubation and mortality when compared to the COVID-19 RT-PCR(−) group. Similarly, higher SARS-CoV-2 viral loads are independently associated with intubation and mortality [19]. A higher viral load in the COVID-19 RT-PCR(+) group could be a possible explanation for the superior intubation and mortality rates in that group.

Other limitations must also be considered. The number of RT-PCR assays performed per patient, a procedure that is known to increase test sensitivity [7], was not reported due to a failure in our database. Nonetheless, according to our institutional protocol, patients with suspected COVID-19 infection but a negative initial RT-PCR result were submitted to at least two other consecutive RT-PCR tests. Therefore, all patients in the RT-PCR(−) group had a minimum of two negative tests. 

Radiological chest CT scan findings were not provided due to a limitation of our database design. Another limitation is the imprecise definition of the criteria used to delimitate the RT-PCR(−) group. Our task force of researchers was instructed to input into our dataset if at discharge the attending physicians considered that the patient had a clinical and radiological presumptive diagnosis of COVID-19, even with a negative SARS-CoV-2 RT-PCR result, after ruling out other differential diagnoses. Still, as the reference public tertiary center in Brazil, we believe that this is reliable information. Our doctors are highly trained to identify suspicious clinical and radiological cases of COVID-19 and to rule-out differential diagnoses. Another limitation is that patients were enrolled from March to August 2020. As of August 2020, in Brazil, mainly the following COVID-19 variants were present: B.1.1.33 (37.8%), B.1.1.28 (32.5%), and B.1.1 (16.4%). The primers used to detect SARS-CoV-2 in our hospital were validated by our Laboratory Division (accredited by the College of American Pathologists) and in line with the circulating SARS-CoV-2 strains in our country. Importantly, the gamma variant is not present in this study as it was only reported in Sao Paulo in January 2021 [20].We do not know whether these findings are applicable for any new variants including alpha, beta, delta, gamma, and omicron.

This study has some strengths too. Many patients were enrolled (*n* = 2998), decreasing the risk of a type II error (stating that there is not an effect or difference when one exists). In terms of data collection, although there were some missing values, only data missing completely at random were observed. Importantly, the results reported in this study with hospitalized and symptomatic patients are not reproducible in an outpatient setting. In conclusion, we found that the presence of at least one positive SARS-CoV-2 RT-PCR result was independently associated with intubation or 28 days death during hospitalization when compared to patients who had only a presumptive clinical and radiological criterion for COVID-19 diagnosis without a second, more probable differential diagnosis. Of note, CT findings are sensitive but not specific for COVID-19 [16], as a result, only patients with both clinical suspicion for COVID-19 and a characteristic CT scan were enrolled in the COVID-19 RT-PCR(−) group. These findings suggest that patients clinically considered to have COVID-19 but with a negative SARS-CoV-2 RT-PCR result may have less risk of COVID-19-related unfavorable outcomes.

## Data Availability

Due to its sensitive nature involving patient health information, the data presented in this study are available upon a reasonable request.

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
