# Peer review of "Distinct Outcomes in COVID-19 Patients with Positive or Negative RT-PCR Test"

_viruses, 2022, doi:10.3390/v14020175_

Round 1
Reviewer 1 Report
The manuscript by Menezes et al. compares COVID-19 patients whose disease was diagnosed by RT-PCR or by other means when the RT-PCR was negative. The authors found that RT-PCR positive patients were more likely to develop severe disease, while COVID-19 patients with a negative RT-PCR result presented a more favorable outcome. This is a nice and well-written study, which included almost 3000 participants. One could argue that patients belonging to the RT-PCR (-) group had in fact another disease and were not infected by the SARS-CoV-2. However, the authors openly discussed their work limitations in the discussion section, which was appreciated.
I have two questions that could contribute to the story:
- What are the diseases that could have been mistaken for COVID-19 ?
- Could we have more details regarding serological analysis (lines 270-272)? Were the patients sampled before, during, or after admission? It would be interesting to know if patients belonging to the PCR negative group were serologically negative when discharged from the hospital (and would question the COVID-19 diagnosis).
Minor corrections:
Flowchart 1: typo. From, clinical.
Line 269-270: the authors wrote twice that CRP values were lower.
Line 272: a parenthesis is missing.
Line 275: why is COVID-19 seropositivity in Sao Paulo relevant in this study?
Lines 342-345: the authors forgot to remove this paragraph
Author Response
Dear Reviewer #1,
Thank you for giving us the opportunity to submit a revised draft of our manuscript titled ''Distinct outcomes in COVID-19 patients with positive or negative RT-PCR test'' to Viruses. We appreciate the time and effort that you have dedicated to providing your valuable feedback on our manuscript. We have been able to incorporate changes to reflect most of the suggestions you provided. We have highlighted the changes within the manuscript. Here is a point-by-point response to your comments and concerns.
1) The manuscript by Menezes et al. compares COVID-19 patients whose disease was diagnosed by RT-PCR or by other means when the RT-PCR was negative. The authors found that RT-PCR positive patients were more likely to develop severe disease, while COVID-19 patients with a negative RT-PCR result presented a more favorable outcome. This is a nice and well-written study, which included almost 3000 participants. One could argue that patients belonging to the RT-PCR (-) group had in fact another disease and were not infected by the SARS-CoV-2. However, the authors openly discussed their work limitations in the discussion section, which was appreciated.
Author response: Thank you for your comment.
I have two questions that could contribute to the story:
2) What are the diseases that could have been mistaken for COVID-19?
Author response: Thank you for your comment. There is a variety of diagnoses that can be mistaken for COVID-19. Fistera et al. reported that, among the differential COVID-19 diagnosis, respiratory diseases are found in 42.7% (114/267) of cases, followed by 14.2% (38/267) infections of other origins, 11.2% (30/267) cardiovascular, 9.0% (24/267) oncological, 6.7% (18/267) gastrointestinal, 4.9% (13/267) urogenital, 1.9% (5/267) neurological, and 9.4% (25/267) miscellaneous diseases. We added the following sentence in the former line 48 ''In this group of patients, the differential diagnosis for COVID-19 should be ruled out. This group of differential diseases includes mainly respiratory diseases, infections of other origins, cardiovascular, oncological, gastrointestinal, urogenital, and neurological diseases [Fistera et al.].''
- Fistera, D., Härtl, A., Pabst, D. et al. What about the others: differential diagnosis of COVID-19 in a German emergency department. BMC Infect Dis 21, 969 (2021). https://doi.org/10.1186/s12879-021-06663-x
3) Could we have more details regarding serological analysis (lines 270-272)? Were the patients sampled before, during, or after admission? It would be interesting to know if patients belonging to the PCR negative group were serologically negative when discharged from the hospital (and would question the COVID-19 diagnosis).
Author response: Thank you. Those details are actually very important. The following sentence was added in the section ''COVID-19 diagnosis'': ''If the patient had more than 7 days of COVID-19 symptoms on hospital admission, COVID-19 serology was also performed. During the inpatient stay, patients that had at least two negative RT-PCR tests but maintained high clinical suspicion for COVID-19 also went through serological testing after, at least, 7 days of symptoms. Unfortunately, due to a shortage of resources, serologic COVID-19 testing could not be done in all eligible patients. COVID-19 serology at discharge was not performed. In every serological testing occasion, both IgM and IgG were tested, and a positive result in either one was considered enough to consider that patients had positive serology status.''
Minor corrections:
4) Flowchart 1: typo. From, clinical.
Author response: Thank you. This issue has been addressed.
5) Line 269-270: the authors wrote twice that CRP values were lower.
Author response: Thank you. The sentence ''On the other hand, CRP values are lower in this group'' has been deleted.
6) Line 272: a parenthesis is missing.
Author response: Thank you. The parenthesis has been added.
7) Line 275: why is COVID-19 seropositivity in Sao Paulo relevant in this study?
Author response: Thank you. We added the COVID-19 seropositivy rate in Sao Paulo in an effort to describe how many patients in our cohort were expected to have a previous positive serology for COVID-19. The following sentences were added in the subsection ''3.2 Laboratory tests''. ''As the majority of our cohort was enrolled before June/2020, less than 11.4% of included patients were expected to have a baseline positive COVID-19 serology''.
8) Lines 342-345: the authors forgot to remove this paragraph
Author response: Thank you. This paragraph has been removed.

Reviewer 2 Report
My major concern is the following:
Line 175-177 and table 1: As the authors mention, the PCR(+) group had a higher prevalence of diabetes, cardiovascular disease, chronic kidney disease peripheral vascular disease and cancer. Therefore, it is not surprising that they saw higher mortality in the PCR(+) group. The authors tried to control for this by applying a multivariate model, but they also acknowledge in the discussion that this may not be enough. The analysis should be redone with a controlled subsample of PCR(+) patients that shows no statistically significant difference with PCR(-) patients in any of these serious co-morbidities that are known to increase the risk of death or intubation for COVID-19 patients.
Line 133: Corman et al describe a protocol based on RdRp, E, N, whereas the authors use E and N.
There should be a discussion on the sensitivity of protocols amplifying two genes vs amplifying three genes, because such a high number of PCR(-) patients in this study implies a problem with the selected protocol. How is this study and its conclusions relevant in late 2021-beginning 2022, where more sensitive RT-PCR protocols are probably being used?
The authors conducted the study between March-August 2020. I think it is worth mentioning that during that time the gamma variant had not been detected yet.
Line 414: It would be a nice addition to the paper to do the following: Go to nextstrain.org, select the latest global analysis – GISAID data and from there select South America and the time-frame of March-August 2020. GISAID will show which lineages were detected during that time. It also shows a frame with the mutations of these genomes. It would be nice to check if the primers of the RT-PCR that the authors used were at coordinates with mutations of those circulating strains.
Line 20: Is RT-PCR the most common test? I thought the self/rapid-tests were the most common ones.
Line 21: do the authors mean negative RT-PCR test results?
Line 27 and 28: RT-PCR(+)
Line 42 and elsewhere: meta-analysis
Line 43: “COVID-19 prevalence”: in what way do the authors mean that? High viral load? Please be more specific. Please also use capitalized COVID-19.
Line 45-46: maybe the authors could expand a bit more on the heterogeneity amongst different tests.
Line 61: in this line the authors should specify again that they discuss patients admitted to hospitals.
Line 90: COVID-19 twice
Line 93: better use study instead of sample.
Line 101: positive twice
Line 110: Please explain that you refer to positive result for another virus.
Flowchart 2 could be incorporated in flowchart 1
Lines 342-345: this part is from the template
Author Response
Dear Reviewer #2,
Thank you for giving us the opportunity to submit a revised draft of our manuscript titled ''Distinct outcomes in COVID-19 patients with positive or negative RT-PCR test'' to Viruses. We appreciate the time and effort that you have dedicated to providing your valuable feedback on our manuscript. We have been able to incorporate changes to reflect most of the suggestions you provided. We have highlighted the changes within the manuscript. Here is a point-by-point response to your comments and concerns.
1) My major concern is the following:
Line 175-177 and table 1: As the authors mention, the PCR(+) group had a higher prevalence of diabetes, cardiovascular disease, chronic kidney disease peripheral vascular disease and cancer. Therefore, it is not surprising that they saw higher mortality in the PCR(+) group. The authors tried to control for this by applying a multivariate model, but they also acknowledge in the discussion that this may not be enough. The analysis should be redone with a controlled subsample of PCR(+) patients that shows no statistically significant difference with PCR(-) patients in any of these serious co-morbidities that are known to increase the risk of death or intubation for COVID-19 patients.
Author response: Thank you for your suggestion. A new analysis with a controlled subsample of patients without diabetes, cardiovascular disease, chronic kidney disease, peripheral vascular disease, cancer, and hematologic malignancy was performed. The following paragraph was added to the ''3.3 Outcomes'' subsection:
''An analysis of a subgroup of patients without comorbidities that differed between the COVID-19 RT-PCR (+) and COVID-19 RT-PCR (-) was also performed. Accordingly, only patients without cardiovascular disease, diabetes, chronic kidney disease, peripheral vascular disease, cancer, and hematologic malignancy were enrolled. In this subgroup, the positive RT-PCR association with intubation (64.4% VS 47.6%, p < 0.001, Table 7) and 28-day mortality (30.8% VS 19.3%, p 0.002, Table 7) was maintained.''
Table 7. Comparison of demographic characteristics, frequency of comorbidities, and outcomes in the COVID-19 RT-PCR (+) and COVID-19 RT-PCR (-) subgroup of patients without cardiovascular disease, diabetes, chronic kidney disease, peripheral vascular disease, cancer, and hematologic malignancy.
|
Variable |
COVID-19 PCR positive n = 595 |
COVID-19 PCR negative n = 207 |
p-value* |
|
Age, years |
62.2 (54.4- 70.9) [ 595] |
61.3 (53.3- 70.7) [ 207] |
0.451 |
|
Male Sex |
356/ 595 (59.8%) |
119/ 207 (57.5%) |
0.555 |
|
Body mass index** |
26.6 (23.3- 31.2) [ 478] |
26.0 (23.5- 30.5) [ 153] |
0.825 |
|
Former or current smoker** |
123/ 594 (20.7%) |
43/ 207 (20.8%) |
0.984 |
|
Current smoker |
55/ 595 ( 9.2%) |
12/ 207 ( 5.8%) |
0.123 |
|
Previous Diseases |
|
|
|
|
Hypertension |
267/ 595 (44.9%) |
90/ 207 (43.5%) |
0.728 |
|
Cerebrovascular disease |
23/ 595 ( 3.9%) |
11/ 207 ( 5.3%) |
0.374 |
|
Arrhythmia** |
14/ 594 ( 2.4%) |
6/ 206 ( 2.9%) |
0.660 |
|
Chronic obstructive pulmonary disease** |
42/ 594 ( 7.1%) |
11/ 207 ( 5.3%) |
0.382 |
|
Asthma |
17/ 595 ( 2.9%) |
7/ 207 ( 3.4%) |
0.703 |
|
End-stage renal disease |
11/ 595 ( 1.8%) |
3/ 207 ( 1.4%) |
0.706 |
|
Rheumatologic disease |
15/ 595 ( 2.5%) |
7/ 207 ( 3.4%) |
0.514 |
|
Liver disease |
14/ 595 ( 2.4%) |
3/ 207 ( 1.4%) |
0.438 |
|
Human immunodeficiency virus infection |
6/ 595 ( 1.0%) |
2/ 207 ( 1.0%) |
0.959 |
|
Outcomes |
|
|
|
|
ICU care |
421/ 595 (70.8%) |
112/ 207 (54.1%) |
<0.001 |
|
Intubation** |
358/ 556 (64.4%) |
81/ 170 (47.6%) |
<0.001 |
|
Days until intubation** |
8.0 (5.0- 12.0) [275] |
9.0 (7.0- 12.0) [50] |
0.153 |
|
Mortality |
218/ 595 (36.6%) |
48/ 207 (23.2%) |
<0.001 |
|
28-Day mortality |
183/ 595 (30.8%) |
40/ 207 (19.3%) |
0.002 |
|
Days until death** |
23.0 (17.0- 30.0) [218] |
23.0 (15.5- 32.8) [46] |
0.845 |
|
Admission until death** |
15.0 (10.0- 22.8) [218] |
12.0 (6.2- 22.5) [46] |
0.190 |
|
Length of stay |
14.0 (7.0- 23.0) [ 595] |
10.0 (5.5- 19.0) [ 207] |
<0.001 |
Variables are expressed as number (%) or median (interquartile range). Bolded values indicated variables with statistically significant associations (significance level of 0.05).
* P values were calculated using the nonparametric Mann-Whitney U command in Python (v 0.24.1), version 15.0, that tests for trend across ordered groups.
** This variable was not assessed in all participants. The denominator is listed next to the variable.
2) Line 133: Corman et al describe a protocol based on RdRp, E, N, whereas the authors use E and N.
There should be a discussion on the sensitivity of protocols amplifying two genes vs amplifying three genes, because such a high number of PCR(-) patients in this study implies a problem with the selected protocol. How is this study and its conclusions relevant in late 2021-beginning 2022, where more sensitive RT-PCR protocols are probably being used?
Author response: Dear reviewer, we are glad that you brought up these topics as this may be a great opportunity for discussion. Although Corman et al described a protocol in which E, followed by RdRp should be used, the laboratory division of our hospital used a protocol with E, followed by N analyzes. In the aforementioned article, the authors attest that “the N gene assay also performed well but was not subjected to intensive further validation because it was slightly less sensitive”. Comparing N and RdRp genomic RNA, Corman et al found that N has a limit of detection of 8.3 RNA copies/reaction, at 95% hit rate [95% CI: 6.1-16.3 RNA copies/reaction], while RdRp’s limit of detection was 3.8 copies/reaction [95% CI: 2.7-7.6]. By these data it is evident that RdRp is superior to N. Nevertheless, we believe that this data suggests that using N as a secondary/confirmatory element following screening with E (first line tool) is still a reliable method that can be useful in a situation in which RdRp is not available. Further information regarding the analyzes involving N can be found in the supplementary materials of Corman et al. In the presenting study, the choice of N over RdRp was a consequence of logistical/resource limitations in our hospital, which made using RdRp primers considerably more difficult when compared to N primers.
We believe that this study and its conclusions are still relevant at the current date because of the relative limitations of resources in most of the BRICS’ health care systems. Although more sensitive RT-PCR protocols may be broadly available in some countries, even wealthier nations such as the USA are struggling with the high demand for tests during the Omicron surge (for instance, see https://www.bostonglobe.com/2021/12/29/nation/long-lines-reflect-states-continued-testing-woes/). Similar scenarios may be found in developing countries whenever the new SARS-CoV-2 variant surge reaches their population. As of the first week of January/2021, Brazil, Russia, and India haven’t reported an increase in cases compatible with what has been seen in several other places; this may suggest that these developing countries are still to be affected by Omicron in the upcoming weeks/months. Even before this new surge, the lack of resources was a challenge for practitioners in those countries. Besides the difficulty in acquiring better tools for diagnosis and/or screening, these professionals may also face situations in which they find themselves forced to make clinical decisions based on the availability of other resources, such as ICU beds, mechanical ventilation, etc. In that sense, the presenting study may serve as a better tool for outcomes prediction and clinical decision making. Therefore, even though more sensitive RT-PCR protocols may be available at the present time, this statement is far from being a guarantee that these tools will be broadly available for health care professionals in countries such as the BRICS’; due to this reality, understanding the differences between patients with positive vs negative RT-PCR test results may help professionals to take better choices in the health care setting. Accordingly, the following paragraph has been added to the discussion section:
“Likewise, another interesting aspect of this study is the relationship between the high number of participants in the PCR(-) group and the RT-PCR protocol used. Corman et al [13] suggested that the first-line protocol should include an RNA analysis of the viral gene E followed by an analysis of the viral gene RdRp. Alternatively, viral gene N could also be used instead of gene RdRp, although it had a higher LOD. In the present study, as it may occur in several other health care institutions throughout other BRICS’ nations, limitation of resources and logistics prevented the laboratory facility from using gene RdRp in the RT-PCR protocol. Due to the difference between the LODs, one may hypothesize that the protocol using gene N contributed to the occurrence of false negatives, increasing the number of participants in the PCR(-) group. However, as mentioned above, we believe that this potential limitation may be present in other institutions, which grants relevance to the presenting results as it mimetizes real daily-care difficulties and may guide practitioners during clinical decision making in the context of limited hospital and/or laboratorial resources (e.g. ICU beds, mechanical ventilation, updated diagnostic protocols, etc).”
3) The authors conducted the study between March-August 2020. I think it is worth mentioning that during that time the gamma variant had not been detected yet. 4) Line 414: It would be a nice addition to the paper to do the following: Go to nextstrain.org, select the latest global analysis – GISAID data and from there select South America and the time-frame of March-August 2020. GISAID will show which lineages were detected during that time. It also shows a frame with the mutations of these genomes. It would be nice to check if the primers of the RT-PCR that the authors used were at coordinates with mutations of those circulating strains.
Author response: Thank you for that suggestion. The nextstrain.org website is really helpful. In Brazil, the FioCruz website imports data from GISAID specifically from Brazil. The following sentences were added in the former line 414: ''As of August 2020 in Brazil, mainly the following COVID-19 variants were present: B.1.1.33 (37.8%), B.1.1.28 (32.5%), and B.1.1 (16.4%) [20]. The primers used to detect SARS-CoV-2 in our hospital were validated by our Laboratory Division (accredited by the College of American Pathologists) and in line with the circulating SARS-CoV-2 strains in our country. Importantly, the gamma variant is not present in this study as it was only reported in Sao Paulo in January 2021. ''
Reference #19: http://www.genomahcov.fiocruz.br/frequencia-das-principais-linhagens-do-sars-cov-2-por-mes-de-amostragem/
4) Line 20: Is RT-PCR the most common test? I thought the self/rapid-tests were the most common ones.
Author response: Thank you for your comment. Indeed, we believe that you are correct. The sentence that stated that RT-PCR is the most common test for diagnosing COVID-19 was replaced for the following: ''Identification of the SARS-CoV-2 virus by RT-PCR from a nasopharyngeal swab sample is the most a common test for diagnosing COVID-19 ''.
5) Line 21: do the authors mean negative RT-PCR test results?
Author response: Thank you. The sentence ''However, some patients present clinical, laboratorial and radiological evidence of COVID-19 infection with negative result(s)'' has been replaced for ''However, some patients present clinical, laboratorial and radiological evidence of COVID-19 infection with negative RT-PCR result(s)''.
6) Line 27 and 28: RT-PCR(+)
Author response: Thank you. This has been corrected.
7) Line 42 and elsewhere: meta-analysis
Author response: Thank you. This has been corrected.
8) Line 43: “COVID-19 prevalence”: in what way do the authors mean that? High viral load? Please be more specific. Please also use capitalized COVID-19.
Author response: Thank you. Indeed the sentence ''A metanalysis published in May 2020 reported a sensitivity of 86% and specificity of 96% for RT-PCR tests; however, only when the Covid-19 prevalence was high'' is confusing. What we were trying to express is that the metanalysis performed by Floriano et al. found that the accuracy of the PCR test for coronavirus diagnosis can change according to the prevalence of the disease. With a prevalence of 50%, common among health professionals with respiratory symptoms, a post-test probability of 96% was found. With a prevalence of 20%, the post-test probability was 84%. With a prevalence of 5%, they found a 55% post-test probability. The following sentence replaced the misleading one: '''A meta-analysis published in May 2020 found that the accuracy of the RT-PCR test for coronavirus diagnosis can change according to the prevalence of COVID-19. With a prevalence of 50%, common among health professionals with respiratory symptoms, a post-test probability of 96% was found. With a prevalence of 20%, the post-test probability was 84%. With a prevalence of 5%, they found a 55% post-test probability.''
9) Line 45-46: maybe the authors could expand a bit more on the heterogeneity amongst different tests.
Author response: Thank you for the suggestion. The review published by Arevalo-Rodrigues et al. in December 2020 enrolled 34 studies (12,057 confirmed COVID-19 cases) and reported that the pooled estimate of false-negative proportion was highly affected by unexplained heterogeneity (tau-squared = 1.39; 90% prediction interval from 0.02 to 0.54). The following sentences were added in the second paragraph of the introduction: ''The heterogeneity is largely unexplained. There are several reasons that can underlie this heterogeneity. Researchers have suggested that these failures in SARS-CoV-2 detection are related to multiple preanalytical and analytical factors, such as lack of standardization for specimen collection, delays or poor storage conditions before arrival in the laboratory, the use of inadequately validated assays, contamination during the procedure, insufficient viral specimens and load, the incubation period of the disease, and the presence of mutations that escape detection''.
10) Line 61: in this line the authors should specify again that they discuss patients admitted to hospitals.
Author response: Thank you. The following sentence ''Therefore, in this study, we explored the specificities of the aforementioned group by comparing 2,292 COVID-19 patients confirmed by SARS-CoV-2 RT-PCR (COVID-19 RT-PCR(+) group) with 706 COVID-19 patients diagnosed by presumptive clinical criteria with negative SARS-CoV-2 RT-PCR results (COVID-19 RT-PCR(-) group)'' has been replaced for ''Therefore, in this study, we explored the specificities of the aforementioned group by comparing 2,292 COVID-19 hospitalized patients confirmed by SARS-CoV-2 RT-PCR (COVID-19 RT-PCR(+) group) with 706 COVID-19 hospitalized patients diagnosed by presumptive clinical criteria with negative SARS-CoV-2 RT-PCR results (COVID-19 RT-PCR(-) group)''.
11) Line 90: COVID-19 twice
Author response: Thank you. This has been corrected.
12) Line 93: better use study instead of sample.
Author response: Thank you. The word sample has been replaced for study.
13) Line 101: positive twice
Author response: Thank you. This has been corrected.
14) Line 110: Please explain that you refer to positive result for another virus.
Author response: Thank you. This has been corrected.
15) Flowchart 2 could be incorporated in flowchart 1
Author response: Thank you. Flowchart 1 was incorporated into flowchart 2.
16) Lines 342-345: this part is from the template
Author response: Thank you. This paragraph has been removed.

Reviewer 3 Report
- Please add the operational definition of COVID-19 cases (both RT-PCR positive as well as RT-PCR negative cases.
- According to IRB agreement, there was waiver of informed consent taking. Please add at the body of manuscript and briefly explain the reasons for waiving the informed consent taking.
- In this study, the authors make comparison between RT-PCR positive and negative. Did the author check any comparison between high and low viral load? It would be interesting. Please add at the revised manuscript.
4.At page 16, line 414, the authors stated that We do not 414 know whether these findings are applicable for the new variants. In this sentence, what variants did you want to say? (Delta or Omicron or etc?) Please clearly describe it.
- The authors stated that this study has many strengths too. Many participants and dtaa are consistent? Can we say those are the strength? Please revise it.
6.At Table-4, the authors described that positivity rate for COVID serology. Please describe clearly, it is IgM or IgG? The authors did not describe anything about serology tests. When did you check serology? Please add at the revised manuscript.
- This study was hospital-based and it can be biased for the asymptomatic cases with RT-PCR positive in the community. There would be many asymptomatic cases with RT-PCR positive. Therefore, please briefly make discussion for those cases at the revised one.
- For RT-PCR negative cases, how many times did you check RT-PCR assay? It was also important for classification of positive and negative cases.
Author Response
Dear Reviewer #3,
Thank you for giving us the opportunity to submit a revised draft of our manuscript titled ''Distinct outcomes in COVID-19 patients with positive or negative RT-PCR test'' to Viruses. We appreciate the time and effort that you have dedicated to providing your valuable feedback on our manuscript. We have been able to incorporate changes to reflect most of the suggestions you provided. We have highlighted the changes within the manuscript. Here is a point-by-point response to your comments and concerns.
1) Please add the operational definition of COVID-19 cases (both RT-PCR positive as well as RT-PCR negative cases.
Author response: Thank you for your suggestion. The following sentence was added to the section ''Materials and Methods'': ''In this research study, the RT-PCR(+) group is formed by patients with a high clinical suspicion of COVID-19 and a positive RT-PCR test for the SARS-CoV-2 whereas the RT-PCR(-) group is formed by patients with a high clinical suspicion of COVID-19, judged by 2 experienced attendings after ruling out differential diagnosis, and at least two negative RT-PCR tests for the SARS-CoV-2.''
2) According to IRB agreement, there was waiver of informed consent taking. Please add at the body of manuscript and briefly explain the reasons for waiving the informed consent taking.
Author response: Thank you for your comment. Indeed, it is very important to justify why there was a waiver of informed consent. The following sentence was added to the ''Informed Consent Statement'' section: ''The informed consent was waived because the signature on the informed consent document would be the only record linking the subject to the research and the principal risk of harm to the subject would be a breach of confidentiality.''
3) In this study, the authors make comparison between RT-PCR positive and negative. Did the author check any comparison between high and low viral load? It would be interesting. Please add at the revised manuscript.
Author response: Thank you for your suggestion. We believe that this discussion is very interesting and could explain, at least in part, the different outcomes between the COVID-19 RT-PCR(+) group and COVID-19 RT-PCR(-) group. Magleby et al. published in June 2020 a very interesting article in Clinical Infectious Diseases. They conducted a retrospective cohort study with 678 COVID-19 patients and compared characteristics and outcomes of patients with high, medium, and low admission viral loads to assess whether viral load was independently associated with intubation and in-hospital mortality. They had 2 main findings:
- Higher viral loads are associated with a higher rate of some comorbidities which may be due to a difficulty to clear the SARS-CoV-2 viral load in this group of patients. In our study, we found that the COVID-19 RT-PCR(+) group when compared to the COVID-19 RT-PCR(-) also had a higher rate of some comorbidities. We have tried to explore this discussion in the 5th paragraph of the ''Discussion'' section:
''Another hypothetical explanation for the higher rate of some comorbidities in the COVID-19 RT-PCR(+) group is a systemic difficulty to clear the SARS-CoV-2 viral load. Our results are in line with previous evidence which demonstrated that COVID-19 patients with cardiovascular disease, chronic kidney disease, active cancer, and hematologic malignancy present a higher viral load on hospital admission and, thus, a higher probability of a positive RT-PCR test [19]. Reasons for higher viral loads specifically in these populations are not fully understood and warrant further investigation. However, in previous studies, chronic obstructive pulmonary disease and age were also associated with higher viral load [18] whereas in the present study they were not different among COVID-19 RT-PCR(+) and COVID-19 RT-PCR(-) groups. This may be caused by the limitations of evaluating only SARS-CoV-2 RT-PCR results.''
- Higher viral loads are independently associated with mortality and intubation. In our study, we also found that a positive SARS-CoV-2 RT-PCR test was independently associated with mortality and intubation in patients with a presumable COVID-19 diagnosis. We added the following paragraph to the discussion:
''Although the COVID-19 RT-PCR(+) group had a higher rate of comorbidities, after an adjustment for the factors that are associated with a worse COVID-19 outcome, the aforementioned group maintained an excessive incidence of intubation and mortality when compared to the COVID-19 RT-PCR(-) group. Similarly, higher SARS-CoV-2 viral loads are independently associated with intubation and mortality [19]. A higher viral load in the COVID-19 RT-PCR(+) group could be a possible explanation for the superior intubation and mortality rates in that group.''
4) At page 16, line 414, the authors stated that We do not know whether these findings are applicable for the new variants. In this sentence, what variants did you want to say? (Delta or Omicron or etc?) Please clearly describe it.
Author response: Thank you for your comment. The following sentence was added explaining which variants were mainly circulating in Brazil until August 2020: ''As of August 2020, in Brazil, mainly the following COVID-19 variants were present: B.1.1.33 (37.8%), B.1.1.28 (32.5%), and B.1.1 (16.4%)''. Except for those, We do not know if our study's findings are applicable to other variants. Accordingly, the following setence was rewritten: ''We do not know whether these findings are applicable for any the new variants including alpha, beta, delta, gamma, and omicron.''
5) The authors stated that this study has many strengths too. Many participants and dtaa are consistent? Can we say those are the strength? Please revise it.
Author response: Thank you. We meant that the magnitude of participants enrolled in this study (n = 2,998) decreases the risk of a type II error (stating that there is not an effect or difference when one exists). We believe that this is a strength. In terms of data collection, although we have some missing values, only missing data completely at random was observed. The sentences ''This study has some strengths too. Many patients were enrolled, and the data are consistent.'' were replaced for ''This study has some strengths too. Many patients were enrolled, and the data are consistent. (n = 2,998) decreasing the risk of a type II error (stating that there is not an effect or difference when one exists). In terms of data collection, although there were some missing values, only missing data completely at random was observed.''
6) At Table-4, the authors described that positivity rate for COVID serology. Please describe clearly, it is IgM or IgG? The authors did not describe anything about serology tests. When did you check serology? Please add at the revised manuscript.
Author response: Thank you. This is actually very important. Serology was checked on admission if the patient had more than 7 days of COVID-19 symptoms and during hospital stay if the patient had at least two negative RT-PCR tests but maintained a high clinical suspicion for COVID-19 and more than 7 days of symptoms. In this context, serology for both IgM and IgG were tested in every testing occasion. Considering the recent background of symptoms and eventual fluctuations in antibodies levels/immunological responses, positive tests for either IgM or IgG were considered enough to count those patients as having positive serology.
The following setence was added in the ''COVID-19 Diagnostics'' subsection to address this issue: ''If the patient had more than 7 days of COVID-19 symptoms on hospital admission, COVID-19 serology was also performed. During the inpatient stay, patients that had at least two negative RT-PCR tests but maintained high clinical suspicion for COVID-19 also went through serological testing after, at least, 7 days of symptoms. Unfortunately, due to a shortage of resources, serologic COVID-19 testing could not be done in all eligible patients. COVID-19 serology at discharge was not performed. In every serological testing occasion, both IgM and IgG were tested, and a positive result in either one was considered enough to consider that patients had positive serology status.''
7) This study was hospital-based and it can be biased for the asymptomatic cases with RT-PCR positive in the community. There would be many asymptomatic cases with RT-PCR positive. Therefore, please briefly make discussion for those cases at the revised one.
Author response: Thank you for the suggestion. In this study, none asymptomatic cases were enrolled. All subjects had some sort of respiratory symptoms (cough, dyspnea) and diffuse infiltrates on X-Rays. However, as this study occurred in a hospital setting, its results are not reproducible in an outpatient scenario were there can be asymptomatic or mildly symptomatic RT-PCR positive patients. The following sentence was added in the last paragraph of the ''Discussion'' subsection: ''Importantly, the results reported in this study with hospitalized and symptomatic patients are not reproducible in an outpatient setting.''.
8) For RT-PCR negative cases, how many times did you check RT-PCR assay? It was also important for classification of positive and negative cases.
Author response: Thank you for the comment. We agree that this is a very important information. Unfortunately, we do not have the exact number of times that the RT-PCR assay was checked. This is an important limitation of our study. We discussed that limitation in the 7th paragraph of the ''Discussion'' subsection:
''Some limitations of our study must be considered. The number of RT-PCR assays performed per patient, a procedure that is known to increase test sensitivity [7], was not reported due to a failure in our database. Nonetheless, according to our institutional protocol, patients with suspected COVID-19 infection but a negative initial RT-PCR result were submitted to at least 2 other consecutive RT-PCR tests. Therefore, all patients in the RT-PCR(-) group had a minimum of two negative tests. ''

Round 2
Reviewer 2 Report
The authors have done a superb job in addressing my concerns.
I was mostly concerned about the differences between the two groups, but the authors have redone this analysis with controled/stratified subsamples and clearly demonstrate that the differences are still statistically significant.
This is a very nice study and I wish the authors all the best with their difficult/challenging job.
As a final note, which needs not be addressed in the manuscript (but just for the author's information, more like a friendly note to them), with the omicron variant there seems to be a complacency in general that this is the end of the pandemic. However, the evolutionary plasticity of coronaviruses and SARS-CoV-2 is very high, thus this complacency may or may not be justified at the end. Nobody knows yet. In my opinion, and assuming the author's close communication with policy makers, there needs to be vigilance, especially at the political level. Please see the latest review (see link bellow) about the evolutionary plasticity of Coronaviruses, the various evolutionary scenarios for SARS-CoV-2 based on observations in several coronaviruses and how easily a few point mutations may change tissue tropism and convert a benign coronavirus in cats into a deadly form (FIPV, in cats again). Let's hope that SARS-CoV-2 does not hold any more surprises.
https://www.mdpi.com/1999-4915/14/1/78/htm
Reviewer 3 Report
I have no more comment.